# TileSparse: Arithmetic-Intensity-Aware Sparse Attention for Compute-Bound LLM Decoding

**Chao Wang** [1]  **Pengfei Zuo** [2]  **Zhangyu Chen** [2]  **Qihui Zhou** [1]  **Tsung-Yi Ho** [1]  **Ming-Chang Yang** [1]

## Abstract

Sparse attention has emerged as a vital technique for long-context inference in Large Language Models (LLMs), effectively accelerating memory-bound decoding by reducing memory access for non-essential keys. However, the assumption that decoding attention is memory-bound has been shattered. The proliferation of Multi-head Latent Attention (MLA) and Multi-Token Prediction (MTP) architectures has effectively rendered the process compute-bound. We observe that, in MLA, Q-heads exhibit a degree of sparsity even when attending to the same key; consequently, traditional sparse attention algorithms introduce significant computational inefficiency in this new regime by rigidly computing interactions between all associated Q-heads and the retrieved keys.

To address this, we propose TileSparse, an arithmetic-intensity-aware (a.i.-aware) algorithm for efficient attention in compute-bound settings. We first introduce a cost model that emphasizes compute budget (compute tile size) rather than memory budget (fetched tokens) when evaluating sparse methods. Next, QK 2D Sparsity prunes unnecessary Q-head–key computations and uses the freed compute to retrieve more semantically important tokens. Because Q-head sparsity differs across keys, we further propose Tiered QK 2D Sparsity and an AutoTuner to choose the best pattern. Experiments show that under tight budgets our method improves accuracy by 40% over state-of-the-art dynamic K-only sparse methods. It also preserves 99% of full-attention accuracy while cutting attention compute by 40.8%, outperforming prior sparse attention approaches.

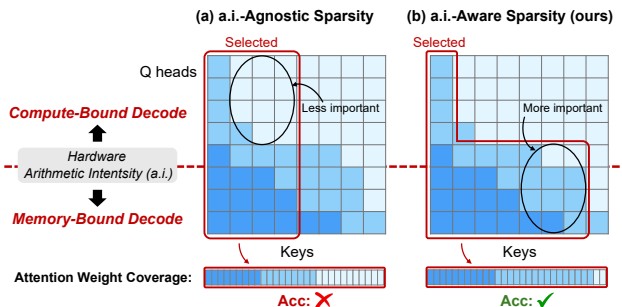

*Figure 1.* Comparison of arithmetic-intensity-agnostic vs. arithmetic-intensity-aware sparsity. The dashed line marks the hardware arithmetic-intensity threshold where decode attention shifts from memory-bound to compute-bound. (a) arithmetic-intensity-agnostic: computes all Q-heads for each selected key, wasting compute on unimportant heads. (b) arithmetic-intensity-aware (ours): selects only important Q-heads and can retrieve more keys under the same compute budget, improving accuracy.

## 1. Introduction

Recently, Large Language Models (LLMs) increasingly support extremely long contexts (e.g., GPT-5, Claude, and DeepSeek) (OpenAI, 2025; Anthropic, 2025; DeepSeek-AI et al., 2025a), enabling applications from long-document understanding to agentic workflows (Zhang et al., 2024b; Goyal & Durrett, 2020; OpenAI, 2022; Zhang et al., 2024a). However, attention scales quadratically with sequence length, creating a major decode-time latency bottleneck (Child et al., 2019; Beltagy et al., 2020). To mitigate this, sparse attention reads only a subset of important tokens, making it a promising solution (Child et al., 2019; Beltagy et al., 2020; Tang et al., 2024; Yuan et al., 2025).

Conventional sparse attention was built on the widely adopted assumption that decode attention is memory-bound (Zhang et al., 2023; Tang et al., 2024; Chen et al., 2024; Yuan et al., 2025), and thus focuses on reducing memory traffic (i.e., fetched tokens). However, memory-efficient techniques such as MLA (DeepSeek-AI et al., 2024; 2025a) and MTP (Qi et al., 2020; Gloeckle et al., 2024), now used in recent models (DeepSeek-AI et al., 2025a; Yang et al., 2025; Team et al., 2025b), increase attention arithmetic intensity and shift decoding toward the compute-bound regime (Zadouri et al., 2025; Williams et al., 2009).

[1]Department of Computer Science and Engineering, The Chinese University of Hong Kong, Hong Kong SAR, China [2]Huawei Technologies Co., Ltd., Shenzhen, Guangdong, China. Correspondence to: Pengfei Zuo <pfzuo.cs@gmail.com>.

*Proceedings of the 43rd International Conference on Machine Learning*, Seoul, South Korea. PMLR 306, 2026. Copyright 2026 by the author(s).

In this regime, conventional designs become inefficient: we observe Q-head sparsity in MLA, yet existing methods still compute *all* query heads for each selected key (Yuan et al., 2025; Chen et al., 2024), wasting compute on unimportant heads (Figure 1).

In this paper, we propose TileSparse, an arithmetic-intensity-aware sparse attention algorithm designed to enable efficient attention computation in compute-bound scenarios. We introduce several key innovations. First, we establish a new cost evaluation paradigm that prioritizes computational budget (compute tile size) over memory access budget (number of fetched tokens) to assess sparse algorithms. Second, we propose QK 2D Sparsity to exploit Q-head sparsity in MLA, moving beyond existing K-only (1D) pruning. It selectively prunes unnecessary Q-head computations for specific keys, reallocating the saved arithmetic budget to retrieve a broader set of semantically critical tokens. Third, we observe that different tokens exhibit different degrees of Q-head sparsity, and propose Tiered QK 2D Sparsity to apply tiered sparsity patterns across tokens. Finally, we introduce an AutoTuner that adapts to the best sparsity pattern automatically.

We evaluate TileSparse on Ruler and LongBench. Under a 256-token budget, TileSparse improves accuracy by 36.4% and 7.7%, respectively, while reducing computational overhead by 18.3% and 40.7% and preserving 99% of full-attention accuracy. Our code is publicly available at https://github.com/ASISys/TileSparse.

In summary, our contributions are as follows:

- We identify the inefficiency issue of applying memory-centric sparse attention to compute-bound MLA architectures.

- We establish a new evaluation paradigm that prioritizes *computational budget*, providing a rigorous framework to assess efficiency in compute-bound scenarios.

- We propose TileSparse, an **arithmetic-intensity-aware framework**. Utilizing QK 2D Sparsity, it transcends traditional K-only (1D) pruning to precisely allocate arithmetic resources.

- We demonstrate that TileSparse achieves superior accuracy-efficiency trade-offs on real-world datasets.

## 2. Related Work

### 2.1. Long-Context LLM

Long-context capability is now a core requirement for LLMs, enabling long-document understanding (Zhang et al., 2024b; Goyal & Durrett, 2020), multi-turn conversations (OpenAI, 2022), and agentic workflows (Zhang

et al., 2024a). Accordingly, both proprietary and open-source models have rapidly expanded context windows, e.g., Claude Sonnet 4.5 (200k) (Anthropic, 2025), GPT-5.2 (400k) (OpenAI, 2025), DeepSeek-V3/R1 and Qwen-3 (128k) (DeepSeek-AI et al., 2025a; Guo et al., 2025; Yang et al., 2025), and Llama 4 Scout (10M) (Meta AI, 2025). However, longer sequences exacerbate attention's quadratic cost and KV-cache footprint, motivating efficient attention optimizations (Child et al., 2019; Beltagy et al., 2020; Yuan et al., 2025; Chen et al., 2024).

### 2.2. Sparse Attention Algorithms

Sparse attention has become a promising solution for accelerating long-context inference by attending to only a subset of important tokens (Child et al., 2019; Beltagy et al., 2020).

Sparse attention can be categorized as *training-based* or *training-free*. Training-based methods (e.g., Native Sparse Attention (Yuan et al., 2025), DeepSeek Sparse Attention (DeepSeek-AI et al., 2025b), and ZigZag (Zhang et al., 2026)) typically require architectural changes and costly re-training, making them hard to retrofit to existing models. In contrast, training-free methods are plug-and-play, leveraging the inherent sparsity of pre-trained attention without gradient updates, which motivates our focus.

Training-free methods can be further divided into *static* and *dynamic* sparsity. Static sparsity uses query-agnostic, position-based patterns (e.g., sliding windows in StreamingLLM (Xiao et al., 2024) and H2O (Zhang et al., 2023)) but may not adapt to changing queries. Dynamic sparsity scores pages and selects the important ones according to the current query token at every decode step, represented by Quest (Tang et al., 2024) and ArkVale (Chen et al., 2024). Both methods follow a three-stage pipeline: *Estimate* a per-page importance score from a compact key digest, *Select* the top-$k$ pages, and run *Core Attention* on the selected pages. They differ in the digest: Quest uses a cuboid-max statistic, while ArkVale adopts a cuboid-mean variant that more faithfully captures aggregate page-level importance.

Finally, dynamic methods must balance selection granularity and hardware efficiency: token-level selection is fine-grained but often causes non-coalesced memory access and poor GPU throughput (Yuan et al., 2025), so recent works (e.g., Quest, ArkVale, NSA) commonly adopt block-level selection to enable memory coalescing and higher efficiency. We also follow this block-level design.

### 2.3. Memory-Efficient Attention

KV-cache memory overhead has long constrained long-context inference, motivating more memory-efficient attention. At first, MHA (Vaswani et al., 2017) (e.g., Llama-

*Figure 2.* Arithmetic intensity of attention across configurations and GPU hardware ridges. The operator becomes compute-bound once the configuration's a.i. exceeds the GPU's ridge.

2-7B (Touvron et al., 2023)) fetches distinct KV vectors per Q head, while GQA (Ainslie et al., 2023) shares KV across groups of Q heads to reduce KV-cache footprint and bandwidth; e.g., Llama-3-70B (64 Q-heads, 8 KV-heads) (Grattafiori et al., 2024) computes $64/8 = 8$ Q heads per KV fetch.

DeepSeek further improves memory efficiency with MLA (DeepSeek-AI et al., 2024; 2025a), which compresses all KV heads into one shared latent representation. In DeepSeek-V3/R1 (DeepSeek-AI et al., 2025a; Guo et al., 2025), one latent KV fetch supports computing all 128 Q heads, dramatically reducing memory bandwidth per query. MLA is adopted by frontier models such as LongCat-Flash (Team et al., 2025b) and Kimi k2 (Team et al., 2025a), and can be extended to arbitrary Transformers (Ji et al., 2025; Meng et al., 2025).

Similarly, Multi-Token Prediction (MTP) (Qi et al., 2020; Gloeckle et al., 2024) predicts $D$ future tokens in parallel, so one KV read supports $D\times$ more Q-head computations and accelerates memory-bound attention. MTP is adopted in models such as DeepSeek-V3/R1, LongCat-Flash (Team et al., 2025b), MiMo-V2-Flash (Team et al., 2026), and Qwen3-Omni (Xu et al., 2025).

## 3. Motivation

Existing sparse attention research has largely operated under the premise that the decode attention is memory-bound. However, as noted in recent hardware-efficiency studies (Zadouri et al., 2025), this foundational assumption is being challenged by the evolution of model architectures. In this section, we analyze how MLA and MTP shift the bottleneck from memory bandwidth to computation, rendering traditional sparse attention methods inefficient.

### 3.1. The Paradigm Shift: From Memory-Bound to Compute-Bound

The bottleneck of the attention operator is determined by its **Arithmetic Intensity (a.i.)**, defined as the ratio of floating-point operations (FLOPs) to memory bytes accessed. Following the general formulation by (Zadouri et al., 2025), under the common BF16/FP16 precision setting, the a.i. during decoding is governed by:

$$\text{a.i.} \approx \frac{2 \cdot g_q}{m_{kv}} \tag{1}$$

where $g_q = H_q/H_{kv}$ is the group size, and $m_{kv} \in \{1, 2\}$ denotes the KV multiplicity (1 for MLA where K and V share a latent, and 2 for MHA/GQA where K and V are distinct). This formula elucidates that a.i. scales linearly with the number of Query heads attending to a single KV head.

In traditional Multi-Head Attention (MHA), where $g_q = 1$ and $m_{kv} = 2$, the a.i. is approximately 1. Even for Grouped Query Attention (GQA) with a typical setting of $g_q = 8$, the a.i. rises to only $\approx 8$. Both are significantly below the arithmetic intensity threshold of modern hardware (e.g., 156 for A100 with FP16/BF16). Consequently, decoding in these architectures is dominated by memory bandwidth, making the computation of head effectively "free."

However, MLA and MTP radically alter this landscape, as illustrated in Figure 2:

1. **MLA High Group Size:** MLA architectures (e.g., DeepSeek-V3) compress KV into a single joint latent head ($m_{kv} = 1$) serving a massive group of Query heads (e.g., $g_q = 128$). Substituting these into Eq. 1 yields an a.i. of $\approx 256$.

2. **MTP Amplification:** Speculative decoding with MTP (e.g., $D = 2$) further multiplies the active Query heads by the number of predicted tokens $D$, pushing the effective a.i. even higher (e.g., $\approx 512$ at $D=2$).

This surge pushes the operation well beyond the hardware ridge of mainstream inference GPUs (e.g., $\approx 156$ on A100 and $\approx 206$ on H200), transitioning the system into a **compute-bound regime**. To verify this theoretical shift, we conducted experiments using the FlashInfer kernel (Ye et al., 2025) across various GPUs (e.g., A100, H20, H200) by varying the number of Q-heads (arithmetic intensity) while maintaining a fixed sequence length. As illustrated in Figure 4, our results show that the attention operator follows the Roofline model:

1. **Memory-Bound Regime:** At low arithmetic intensity (where the number of Q-heads is small, such as in GQA), the operator is memory-bound. This is indicated by the hollow points in Figure 4, where execution time remains nearly constant regardless of head count.

2. **Compute-Bound Regime:** Once the number of Q-heads increases beyond the hardware's ridge point (characteristic of MLA architectures), the system enters a compute-bound regime. In this phase, the execution time increases linearly with the number of heads ($R^2 > 0.97$).

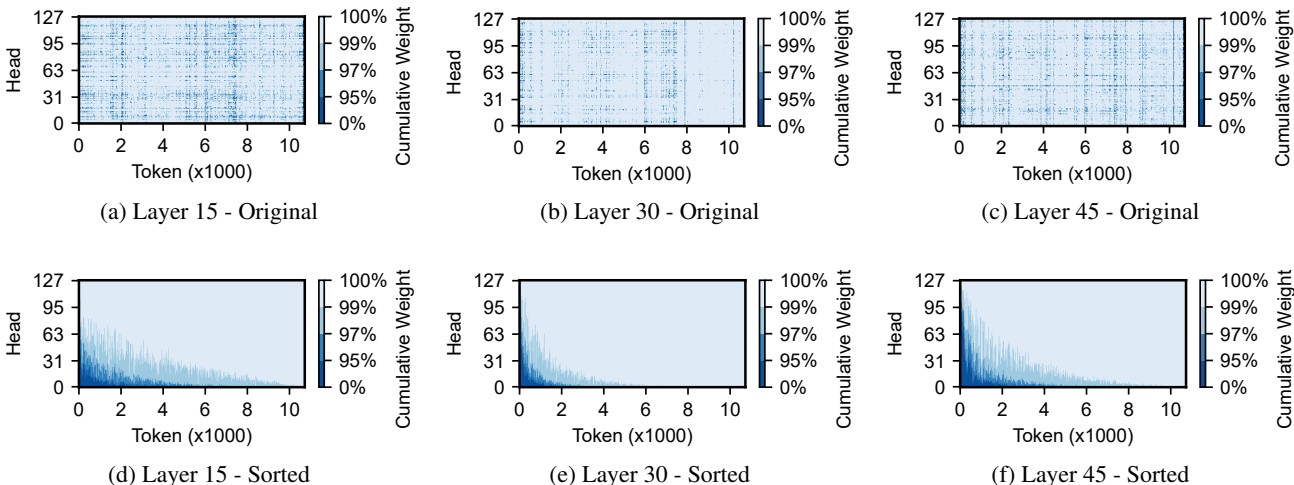

Figure 3. Attention weight heatmaps for a *gov_report* example (LongBench) at Layers 15, 30, and 45. Each cell shows the attention weight $a_{h,t} = \text{softmax}(QK^\top/\sqrt{d})_{h,t}$ assigned by Q-head $h$ to token $t$; darker = heavier weight. Top row (a)–(c): raw maps in original (Q-head, token) order. Bottom row (d)–(f): tokens sorted by descending importance and, per column, Q-heads sorted by descending contribution. Shading bands stratify cells by the cumulative-mass prefix they fall into: dark = top 95%, light = marginal 99% → 100% tail.

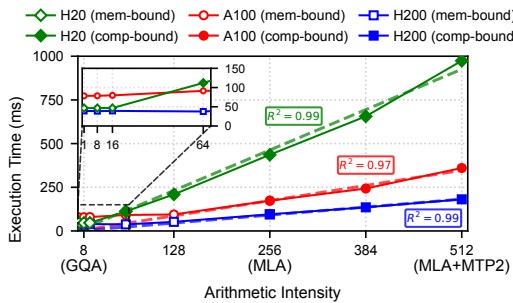

Figure 4. Empirical validation of the piecewise-linear Roofline behavior of attention decoding on A100, H20, and H200. As arithmetic intensity grows, execution time stays flat in the memory-bound regime (hollow markers, see inset) and then grows linearly once a.i. crosses the ridge point ($R^2 > 0.97$, fitted only over the compute-bound segment).

These findings confirm that in modern architectures like MLA and MTP, **increasing the number of attention heads is no longer "free."** Consequently, computation efficiency becomes a critical factor that must be optimized alongside memory access.

### 3.2. The Inefficiency of K-Only Sparsity

This shift invalidates the core assumption of traditional "token-retrieve" sparse attention methods (e.g., ArkVale). These methods focus solely on filtering the Key ($K$) dimension, assuming that once a Key is retrieved, computing it against all Query heads is negligible.

In the compute-bound regime of MLA, this assumption fails. With an a.i. of 256, the massive Q-head computation be-

comes the bottleneck. Blindly calculating interactions for all Q-heads against retrieved Keys—regardless of their semantic relevance—wastes significant computational resources. Therefore, to achieve high resource efficiency in modern LLMs, sparsity must be applied not just to memory access (Keys), but explicitly to computation (Query heads).

### 3.3. Unveiling Q-Head Sparsity in MLA

To explore opportunities for improving computational efficiency, we investigate Q-head sparsity in MLA. Let $a_{h,t} = \text{softmax}(QK^\top/\sqrt{d})_{h,t}$ denote the softmax attention weight that Q-head $h$ assigns to past token $t$ at a decoding step; the weights are non-negative and sum to one, and each $a_{h,t}$ is widely used as a proxy for that token's importance (Tang et al., 2024; Chen et al., 2024). Capturing more cumulative attention mass typically means retrieving more historical information and thus higher accuracy. To quantify how much of the attention is concentrated in how few head–token entries, we sort $\{a_{h,t}\}$ in descending order and take the smallest prefix whose sum reaches $p$ as the *top-$p$ cumulative-mass set*; entries outside this set form the marginal $p \to 100\%$ tail. Figures 3(a)–(c) mark the top-95% set on the raw maps at Layers 15, 30, and 45. Figures 3(d)–(f) further reorder the same heatmaps by importance and stratify cells by their cumulative-mass band.

**Observation 1: Significant Head-Level Sparsity in MLA.** The sorted heatmaps exhibit pronounced sparsity across attention heads for the vast majority of tokens. This finding exposes a fundamental inefficiency in traditional *Key-only* sparse attention algorithms: they typically identify important Keys but compute scores for **all** associated Q-heads.

Our visualization proves that full-head computation is unnecessary, as a large portion of (Q-head, Token) entries fall into the light-blue $99\% \rightarrow 100\%$ tail and contribute almost no probability mass to the output. Operating at the granularity of individual Head-Token pairs is thus essential to reclaim the wasted computational budget.

**Observation 2: Contour-like Distribution of Importance.** A distinct structural pattern emerges where the density of activated heads scales proportionally with token importance. The high-mass entries (dark-blue band) form a *contour-like* distribution (or waterfall pattern). Specifically, highly significant tokens (left side of the x-axis) engage a larger proportion of attention heads, while less significant tokens rely on only a few specific heads. This structured distribution suggests that head-level activation can be dynamically adjusted based on the token's cumulative importance.

To our knowledge, we are the first to observe **within-group Q-head sparsity**, i.e., sparsity *inside* a single head group consisting of one KV head and its associated Q-heads. Prior works (Voita et al., 2019; Michel et al., 2019; Takbir et al., 2025; Xiao et al., 2025; Jin et al., 2025; Shrestha et al., 2025; Lin et al., 2026) have explored head-wise sparsity, but only at the group level — a KV head together with all of its associated Q-heads is treated as a single indivisible unit. They had little incentive to look inside a group because MHA and GQA are memory-bound, so computing every Q-head in a group is effectively free and can only increase the captured attention mass. Our work targets the compute-bound MLA regime, where pruning at this finer Q-head-in-group granularity directly reduces arithmetic intensity and reclaims ineffective computation. A detailed comparison between our work and recent head-wise sparsity designs is provided in Appendix C.

## 4. Design

We propose **TileSparse**, an arithmetic-intensity-aware sparse attention framework for compute-bound decoding (e.g., MLA). We first define an arithmetic-intensity-aware cost metric, then introduce **QK 2D Sparsity** to extend sparsity to the query dimension. We further propose **Tiered QK 2D Sparsity** and an **AutoTuner** to preserve accuracy under high sparsity.

### 4.1. Arithmetic Intensity-Aware Formulation

As analyzed in Section 3, the transition to architectures like MLA has shifted the system bottleneck from memory IO to arithmetic throughput. While memory-bound sparse algorithms typically use the number of fetched pages as a cost metric, we propose the **Number of Active Compute Entries** as the primary metric for compute-bound scenarios. Active entries represent the chosen

$(query\_head, key\_token)$ pairs in the attention matrix that are selected for computation.

In our formulation, we operate on the granularity of KV pages rather than individual tokens to ensure memory access coalescing, a design choice discussed in Section 2. Let $\mathcal{P}$ be the set of $P$ total available KV pages in the context, and $H$ be the total number of attention heads. The total compute cost $C_{total}$ is proportional to the sum of active query heads across all processed pages:

$$C_{total} \propto \sum_{p \in \mathcal{P}} |\mathcal{H}_p| \qquad (2)$$

where $\mathcal{H}_p \subseteq \{1, \ldots, H\}$ is the subset of query heads assigned to a specific page $p$. To ensure hardware efficiency, we enforce a global compute budget $B$. Following the principles of the Roofline model, we identify the **Ridge Point** $\tau_{hw}$ as the critical threshold for arithmetic intensity. To avoid efficiency degradation, we introduce a hardware-aware constraint:

$$|\mathcal{H}_p| \in \{0\} \cup \left[\frac{\tau_{hw} \cdot m_{kv}}{2}, H\right] \qquad (3)$$

Here, $m_{kv} = 1$ for MLA, and $m_{kv} = 2$ for architectures where K and V do not share a single latent vector (consistent with Eq. (1)). If $|\mathcal{H}_p|$ is non-zero but falls below the ridge point $\frac{\tau_{hw} \cdot m_{kv}}{2}$, the execution remains memory-bound. In this case, the kernel execution time does not decrease, but the captured information decreases compared to a configuration that maintains $|\mathcal{H}_p|$ exactly at the ridge point. Thus, a page must either be fully pruned or computed with sufficient heads to saturate the hardware's compute units.

### 4.2. QK 2D Sparsity

Traditional sparse methods primarily focus on K-only sparsity ($|\mathcal{H}_p| = H$). Our proposed **QK 2D Sparsity** introduces a fine-grained 2D selection mechanism controlled by two primary parameters: the page-level budget $N_p$ and the head-level budget $N_h$.

1. **Estimation** (following ArkVale): this stage estimates the page-granularity distribution of attention importance shown in Figure 3. To this end, each page is summarized by a compact digest, computed once when the page is first filled by the KV cache and cached for future estimation. The digest of a page comprises only a max-vector and a min-vector, so its footprint is far smaller than that of the original keys. At each decode step, an element-wise product between the query and the max/min vectors yields the per-page importance score, namely an upper bound on the maximum query–key dot product within the page; ArkVale further refines this bound with a mean variant that gives a tighter estimate of aggregate page-level importance

(see ArkVale (Chen et al., 2024) for the formal expression). The digest is architecture-agnostic toward latent structures such as MLA, and the resulting scores are highly correlated with true attention mass (Tang et al., 2024; Chen et al., 2024).

2. **Page Selection** (following ArkVale): Based on the importance scores, a candidate list of the most vital pages is selected by topk, providing the foundation for subsequent fine-grained selection.

3. **Q-Head Selection**: As the core of our proposed design, the system generates a dynamic Q-head mask vector for each selected page. By analyzing the predicted scores of a specific page across different query heads, this step precisely identifies the heads with the highest contribution to that particular page, thereby constructing a sparse mapping between heads and pages. This fine-grained selection mechanism ensures that computational resources are concentrated only on meaningful pairs, effectively eliminating the overhead caused by redundant query heads.

4. **Core Attention Execution**: The final attention computation is performed exclusively for the pairs specified by the selected page list and their corresponding Q-head masks. Pairs that are not selected are skipped entirely, with their attention weights implicitly treated as zero.

This 2D sparsity approach offers two distinct advantages. First, under a fixed compute budget $B$, our design can retrieve a significantly larger number of pages by pruning redundant heads, thereby capturing more semantic information and improving model accuracy. Alternatively, to achieve the same level of information recall as K-only methods, our algorithm can operate with a much lower compute budget, leading to substantial latency reductions in compute-bound regimes.

### 4.3. Tiered QK 2D Sparsity

While QK 2D Sparsity enables flexible budget allocation, naively applying uniform sparsity across all pages leads to accuracy degradation. Prior work has observed that KV pages exhibit a long-tail importance distribution: a few "critical" pages contribute significantly to model precision, while the majority of "ambient" pages provide supplementary context.

To address this, we propose **Tiered QK 2D Sparsity**. We divide the sorted KV pages into a sequence of tiers, where each tier $m$ is assigned both a page budget $count_m$ and a query head budget $h_m$. This mechanism allows us to allocate higher computational density to the most influential

tokens while maintaining a broad context through aggressive sparsity in the tail tiers.

Specifically, we utilize a two-tier configuration comprising: (1) a **Critical Tier**, where $\alpha$ pages are processed with 100% of the query heads ($H$) to protect the most vital semantic information; and (2) an **Ambient Tier**, where $\beta$ pages are computed using only $h$ heads to capture supplementary context efficiently, as illustrated in Figure 1.

The number of tiers itself is a design choice that trades algorithmic optimality against system overhead. More tiers let the selected compute region trace the contour-like importance distribution in Figure 3 more closely. However, since each tier uses a distinct active head count that is typically fixed at kernel launch, additional tiers cannot share one launch and incur extra launch overhead as well as a larger configuration search space, both of which add system complexity. Experiments show that two tiers already capture the bulk of the contour structure while keeping the kernel pipeline lean; richer hierarchies, ideally implemented via variable-size or work-stealing kernels, are left to future work.

---

**Algorithm 1** Tiered QK 2D Sparsity Execution

---

**Input:** Query $Q$, KV Digests $\mathcal{D}$, Tier Configuration List $L = \{(count_1, h_1), (count_2, h_2), \dots\}$
**Output:** Attention Output $O$
$S \in \mathbb{R}^{H \times P} \leftarrow \text{ComputeProxyScores}(Q, \mathcal{D})$
$S_{page} \in \mathbb{R}^P \leftarrow \sum_{h=1}^{H} S[h, :]$
$Idx_{sorted} \leftarrow \text{Argsort}(S_{page}, \text{descending})$
Initialize binary execution mask $M \in \{0, 1\}^{H \times P} \leftarrow 0$
**for** each tier $(count_m, h_m) \in L$ **do**
    $\mathcal{P}_m \leftarrow$ Select next $count_m$ pages from $Idx_{sorted}$
    **for** each page $p \in \mathcal{P}_m$ **do**
        $\Omega_{heads} \leftarrow$ Indices of top $h_m$ scores in $S[:, p]$
        $M[\Omega_{heads}, p] \leftarrow 1$
    **end for**
**end for**
$O \leftarrow \text{SparseAttentionKernel}(Q, K, V, M)$
**return** $O$

---

### 4.4. AutoTuner

The optimal sparsity configuration is sensitive to the distinct attention distributions across different model layers and the sparsity characteristics of the dataset. While manual offline tuning can identify near-optimal parameters by exhaustively searching for the highest precision, this process is prohibitively slow and fails to generalize to varying runtime workloads.

To address this, we introduce an online dynamic AutoTuner. The tuner selects the most effective configuration from a set of pre-defined candidate schemes by maximizing the

**Estimated Attention Mass Coverage**. For each candidate scheme, the tuner estimates the total attention weight it would capture within the $H \times P$ importance score matrix. By choosing the scheme with the highest coverage ratio, the system adaptively optimizes its sparsity patterns to the current semantic context. This tuning process is performed periodically, ensuring high accuracy across diverse prompts with minimal computational overhead.

## 5. Experiments

### 5.1. Setting

We evaluate our design on various common long-context benchmarks, including four representative tasks from **RULER-16K** (NIAH, CWE, FWE, and VT) (Hsieh et al., 2024) and all 21 datasets from **LongBench** (Bai et al., 2024). We employ the widely-used **DeepSeek-V3-0324** as our evaluation backbone. We compare our design against **Full Attention** and the state-of-the-art **K-only sparsity** method ArkVale (Chen et al., 2024).

Following the methodology of ArkVale, for all experimental configurations, we do not apply sparsity to the first two layers of the model. Additionally, the last two pages (page size = 32) of the context are always selected as local context to preserve immediate semantic coherence.

We target the **NVIDIA A100** ($\tau_{hw}$=156) as the reference hardware throughout this study (we discuss other target hardware such as H200 in Appendix A). Accordingly, the Auto-Tuner search space is configured for A100: we use a two-tier Tiered QK 2D Sparsity, where the **Critical Tier** sweeps a budget ratio $x \in \{12.5\%, 25\%, 37.5\%, 50\%, 75\%, 100\%\}$, and the **Ambient Tier** uses the remaining $(1-x)$ budget with $h \in \{80, 96, 112\}$ heads — satisfying the A100 efficiency constraint $h \geq \tau_{hw}/2 = 78$ (for $m_{kv}$=1). The Cartesian product of $x$ and $h$ forms the candidate configuration set for our AutoTuner. Attention latency measurements in Section 5.5 are likewise reported on A100 with vLLM (v0.13.0). For accuracy evaluation, since the sparse configuration alone determines the attention output regardless of the executing GPU, we run it on a single NVIDIA H200 node, which fits DeepSeek-V3-0324 in one machine to facilitate accuracy evaluation.

### 5.2. Results on RULER

In this section, we analyze the accuracy improvement brought by TileSparse under the **same computational budget**. The horizontal axis represents the *Equivalent Token Budget*, normalizing the compute entry budget into the number of tokens retrieved if all heads were active.

**Needle In A Haystack (NIAH).** Figure 5 evaluates NIAH accuracy where difficulty is controlled by $V$ (needles per

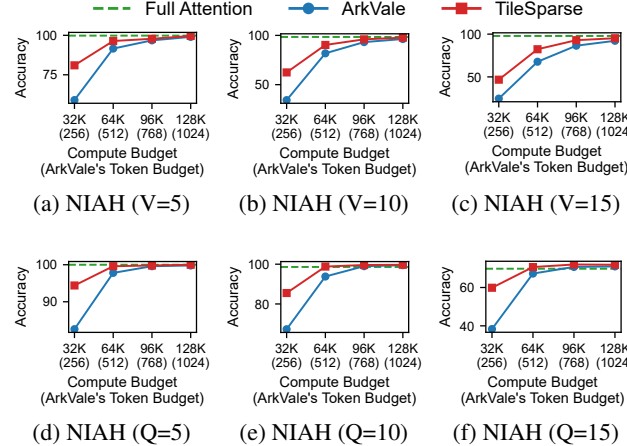

*Figure 5.* **RULER NIAH Accuracy.** Performance with varying needle counts ($V$) and query counts ($Q$). TileSparse maintains high accuracy by reallocating budget to cover more key pages via Q-head sparsity.

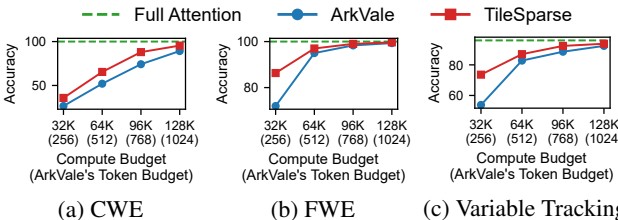

*Figure 6.* **Other RULER Tasks.** Accuracy on global statistics (CWE/FWE) and multi-hop state tracking (VT).

query) and $Q$ (total queries). These parameters control the required context length retrieval and the density of information.

TileSparse consistently outperforms ArkVale across all scenarios. While ArkVale is forced to discard entire KV pages to stay within budget, our 2D design prunes redundant heads to accommodate more pages. This expanded coverage ensures that multiple needles are more likely to be retrieved, leading to a $40\%$ accuracy gain in low-budget settings where context length is the primary bottleneck.

**Synthetic Tasks.** As shown in Figure 6, global tasks like CWE/FWE and multi-hop tasks like VT require broad context access. Our method outperforms 1D sparse baselines by maintaining a longer effective context window, proving that TileSparse is inherently more efficient for compute-bound LLM architectures.

### 5.3. Results on LongBench

We evaluate real-world performance on LongBench. We report the average accuracy across all 21 datasets and showcase results for five representative subsets (Gov Report, MultiFieldQA-en, Qasper, LSHT, and SAMSum).

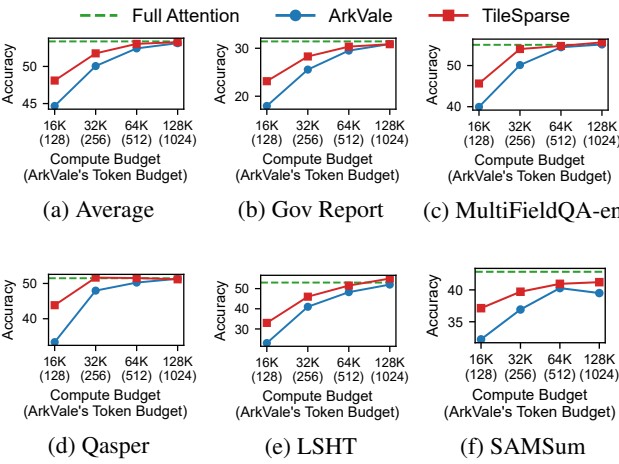

(a) Average  (b) Gov Report  (c) MultiFieldQA-en

(d) Qasper  (e) LSHT  (f) SAMSum

*Figure 7.* **LongBench Accuracy.** Evaluation across diverse real-world tasks.

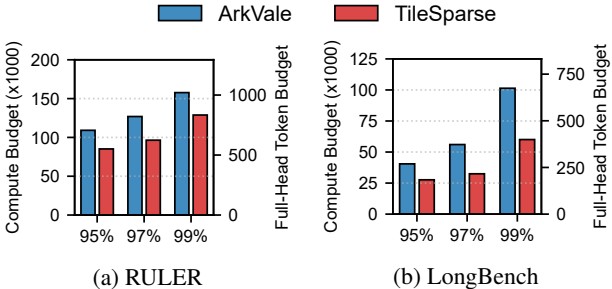

(a) RULER  (b) LongBench

*Figure 8.* **Budget Efficiency.** Budget required to reach various accuracy thresholds.

TileSparse achieves significantly higher scores at low compute budgets. By preserving essential semantic anchors via 2D allocation, TileSparse reaches accuracy levels comparable to Full Attention much faster than ArkVale, highlighting its practical serving efficiency.

### 5.4. Budget Efficiency Analysis

We analyze the compute budget required to reach specific accuracy targets relative to Full Attention performance.

As shown in Figure 8, TileSparse enables substantial budget reduction. To achieve 99% of Full Attention accuracy, our design reduces the budget by approximately **18.3%** on RULER and **40.8%** on LongBench compared to K-only sparsity. This validates that 2D pruning is more resource-efficient than aggressive KV eviction in compute-bound regimes.

### 5.5. Latency Analysis

We evaluate the end-to-end performance of the attention operators within **TileSparse** compared to Full Attention and ArkVale, maintaining a 99% accuracy threshold relative to

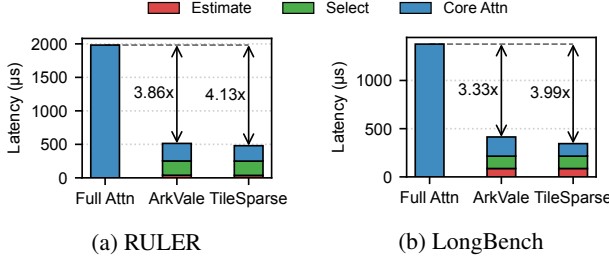

(a) RULER  (b) LongBench

*Figure 9.* **Latency Comparison.** End-to-end inference latency of different designs relative to Full Attention under a 99% accuracy constraint.

the Full Attention baseline. Our implementation is built upon vLLM (v0.13.0) by modifying the Triton MLA operator to support sparse computation. All latency benchmarks are conducted on NVIDIA A100 GPUs.

As illustrated in Figure 9, TileSparse consistently achieves higher speedups than ArkVale relative to Full Attention across both benchmarks. Compared to ArkVale, TileSparse achieves a reduction in core attention latency of **12.2%** and **34.7%** on the RULER and LongBench datasets, respectively. These improvements closely align with the theoretical budget reductions of **18.7%** and **40.8%**. End-to-end, the TileSparse attention operator reaches **4.13×** on RULER and **3.99×** on LongBench over Full Attention, versus ArkVale's **3.86×** and **3.33×** — a **7.0%** and **19.8%** wall-time gain, respectively.

Furthermore, our profiling results demonstrate that the overhead of the AutoTuner is negligible in practice. Each AutoTuner invocation approximately doubles the attention latency of the step it fires on, so the amortized overhead is roughly $1/N$ of total decode compute when the tuner is triggered every $N$ steps. At our default $N{=}10{,}000$, this amounts to only $\sim 0.01\%$; even an aggressive $N{=}600$ (typically less than a minute of decoding) to track rapidly varying workloads still costs only $\sim 0.17\%$, leaving wide headroom between responsiveness and amortized cost.

### 5.6. Ablation Study

To further understand the efficacy of our components, we conduct an ablation study using the RULER benchmark. Each data point in Figure 10 represents a static configuration under a fixed budget of 65,536 entries (512 equivalent full-head tokens).

The x-axis denotes the number of critical tokens processed with 100% Q-heads. A count of 512 represents **K-only sparsity**, while 0 represents **single-tier QK sparsity**. Our analysis reveals several key insights:

(1) **Inverted U-Shape Performance and Multi-tier Necessity**: Model accuracy exhibits a distinct inverted U-shape

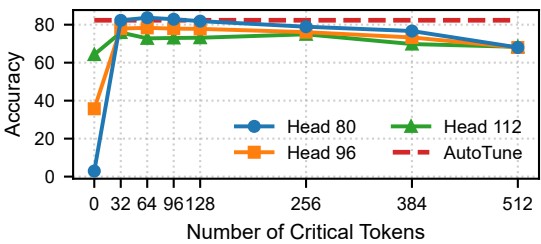

*Figure 10.* **Ablation Study on Tiered QK Sparsity.** Each point is a single static (model-wide, all layers) configuration evaluated on RULER under a fixed budget of 65,536 entries (512 equivalent full-head tokens). **X-axis:** number of *critical* tokens computed with 100% Q-heads ($x=512$ corresponds to K-only sparsity, $x=0$ corresponds to single-tier QK sparsity). **Y-axis:** accuracy. Each **solid curve** sweeps $x$ at a fixed *ambient-tier* Q-head count (i.e., the number of Q-heads used for the non-critical tokens); the **dashed line** is the dynamic AutoTuner, which selects the configuration adaptively at runtime.

distribution relative to the number of critical tokens. The significant performance leap of tiered configurations over the single-tier baseline ($x = 0$) proves that a multi-tier strategy is essential to protect pivotal information. Furthermore, the existence of a clear accuracy peak validates that there is an optimal balance between preserving full-head precision for heavy-hitters and reallocating budget to expand overall context coverage.

(2) **Superiority over K-only**: All peak tiered configurations substantially outperform the K-only baseline ($x = 512$). This demonstrates that in compute-bound scenarios, the fine-grained reallocation of the FLOPs budget via QK 2D Sparsity is fundamentally more efficient than traditional K-only sparsity.

(3) **Effectiveness of the AutoTuner**: The dynamic Auto-Tuner mechanism (dashed red line) consistently tracks the peaks of the static configuration curves across different ambient head counts. This confirms the algorithm's ability to adaptively identify near-optimal sparsity patterns for varying runtime semantic contexts without manual tuning.

## 6. Conclusion

We revisit sparse attention in the compute-bound decoding regime (e.g., MLA) and propose TileSparse, an arithmetic-intensity-aware framework that budgets attention by active compute entries and exploits query-head sparsity via QK 2D Sparsity. Tiered QK 2D Sparsity and an online AutoTuner further balance compute density across pages to preserve accuracy. Experiments on RULER and LongBench show that TileSparse achieves a better accuracy–efficiency trade-off than state-of-the-art K-only sparsity, approaching full-attention quality with substantially less attention compute.

## Impact Statement

This paper presents work whose goal is to advance the field of Machine Learning. There are many potential societal consequences of our work, none which we feel must be specifically highlighted here.

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

| | MHA (a.i. ≈ 1) | GQA (a.i. ≈ 8) | MLA (a.i. ≈ 256) | MLA+MTP2 (a.i. ≈ 512) | MLA+MTP3 (a.i. ≈ 768) | MLA+MTP4 (a.i. ≈ 1024) |
|---|---|---|---|---|---|---|
| H20 (a.i. ≈ 37) | 0.03 | 0.22 | 6.92 | 13.84 | 20.76 | 27.68 |
| A100 (a.i. ≈ 156) | 0.01 | 0.05 | 1.64 | 3.28 | 4.92 | 6.56 |
| H200 (a.i. ≈ 206) | 0.00 | 0.04 | 1.24 | 2.49 | 3.73 | 4.97 |
| B200 (a.i. ≈ 281) | 0.00 | 0.03 | 0.91 | 1.82 | 2.73 | 3.64 |

*Figure 11.* Arithmetic intensity of attention against B200's hardware ridge ($\tau_{hw} \approx 281$). Vanilla MLA (DeepSeek-V3, a.i.≈256) sits just below the ridge and remains memory-bound; pairing with MTP-2 (a.i.≈512) crosses the ridge into the compute-bound regime.

## A. Hardware Generalization Beyond A100

The *leverage* of arithmetic-intensity-aware sparsity stems from the freedom to choose the ambient-tier Q-head count $h$ subject to $h \geq \tau_{hw}/2$ (for $m_{kv}=1$): a lower $\tau_{hw}$ admits a wider feasible range for $h$, exposing more configurations among which the AutoTuner can trade compute for additional pages. On A100 ($\tau_{hw}\approx156$, $h \geq 78$), this range spans the full ablation in Figure 10, where smaller ambient $h$ consistently lifts the accuracy peak. On H200 ($\tau_{hw}\approx206$, $h \geq 103$), the higher bound pushes the feasible range toward $h=128$, leaving little room for the ambient tier to differ from the critical tier — the regime where arithmetic-intensity-aware sparsity, by construction, diminishes. Concretely, in Figure 10 only $h=112$ remains feasible on H200; the shrunken search space lowers the attainable accuracy peak, yet it still surpasses the K-only baseline at $x=512$.

The shrinking design-level leverage on H200 reflects a hardware trend: newer GPUs steadily lift the ridge point $\tau_{hw}$ — the ratio of peak FP16/BF16 throughput to HBM bandwidth — which rises to $\tau_{hw}\approx281$ on B200, substantially higher than A100's $\approx 156$ and H200's $\approx 206$ (Figure 11). Plugging the DeepSeek-V3 configuration ($g_q=128$, $m_{kv}=1$) into Eq. 1 yields a.i.≈256, which sits just *below* B200's ridge and therefore remains memory-bound. However, pairing MLA with MTP-$D$ amplifies a.i. by a factor of $D$ (Section 3.1) and scales the feasible range of $h$ back up: under MTP-2 the a.i. rises to $\approx 512$ and the operator re-enters the compute-bound regime on both H200 and B200, restoring the arithmetic-intensity-aware leverage where TileSparse's Q-head-aware sparsity provides direct benefits. We note that translating theoretical speedups into wall-time gains becomes harder on newer hardware: B200's substantially larger SM count amplifies kernel tail effects, calling for variable-size or work-stealing kernels, which we leave to future work.

## B. Comparison with Additional Baselines

The main paper compares TileSparse against ArkVale, the strongest existing training-free dynamic K-sparsity baseline for MLA. To assess robustness along additional axes, we further benchmark against two baselines that span both static

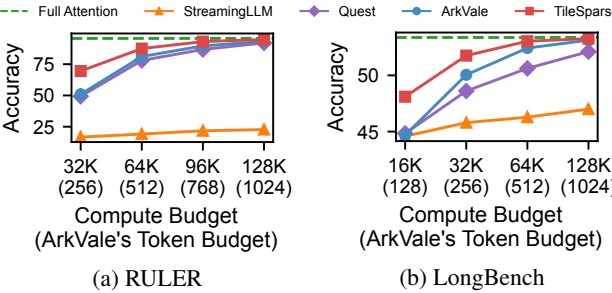

(a) RULER  (b) LongBench

*Figure 12.* Average accuracy on RULER and LongBench across the expanded baseline set.

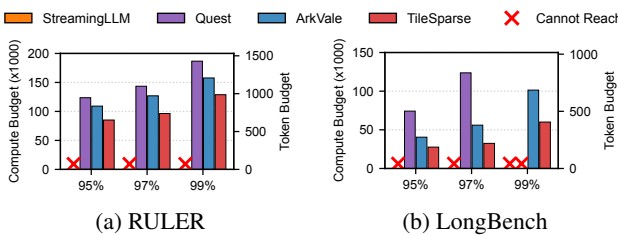

(a) RULER  (b) LongBench

*Figure 13.* Budget required to reach a target fraction of Full Attention accuracy. The cross markers indicate cases where a baseline fails to reach the target even when granted three times the budget TileSparse requires.

and dynamic sparsity: StreamingLLM (Xiao et al., 2024), a representative fixed-window static-sparsity method, and Quest (Tang et al., 2024), a query-aware dynamic K-sparsity method using a cuboid-max digest.

Figure 12 reports the average accuracy on RULER and LongBench as the per-token compute budget varies. TileSparse consistently outperforms StreamingLLM, Quest, and ArkVale on both benchmarks and across the entire budget range. The performance gap is consistent with the underlying design choices of the two added baselines: StreamingLLM lacks query-dependent dynamic selection and therefore misses critical long-range semantic anchors, while Quest's cuboid-max digest is driven by token-level extreme values and is less faithful at capturing aggregate page-level semantic importance than the cuboid-mean digest adopted in ArkVale and TileSparse.

Figure 13 re-examines the same comparison from the budget-efficiency perspective. TileSparse reaches every accuracy target with the lowest budget on both RULER and LongBench, and the advantage widens at the strictest targets.

## C. Comparison with Head-wise Sparsity Designs

This appendix elaborates on how our work differs from recent head-wise sparsity designs, as briefly noted in Sec-

*Table 1.* Comparison of head-wise sparse attention designs. Recall that $g_q = H_q/H_{kv}$ (Section 3.1).

| Method | Problem Scenario | Pruning Unit | $g_q$ | Selection Criterion |
|---|---|---|---|---|
| Quest / ArkVale | Memory-bound | Group | Maximized | Attention weight |
| DuoAttention / LycheeDecode | Memory-bound | Group | Maximized | Trained classifier |
| MoH / Polar Sparsity | Memory-bound | Group | Maximized | Trained router |
| **TileSparse (Ours)** | **Compute-bound** | **Q-head in Group** | **Adaptive, a.i.-aware** | **Attention weight** |

tion 3.3. We first summarize the most relevant works, then compare their design choices against ours in Table 1.

Quest (Tang et al., 2024) performs query-aware K-sparsity by selecting important KV tokens based on token-level extreme values (cuboid-max). DuoAttention (Xiao et al., 2025) partitions heads into "retrieval heads" with full KV cache and "streaming heads" with constant-size cache via gate optimization. LycheeDecode (Lin et al., 2026) uses a HardKuma-based mechanism to partition attention heads into retrieval heads (for identifying crucial tokens) and sparse heads (which reuse the identified crucial tokens). MoH (Jin et al., 2025) treats individual KV heads as experts and uses a router to activate Top-$K$ KV heads for each token, thus reducing KV-cache fetching. Polar Sparsity (Shrestha et al., 2025) employs Selective Group Attention for GQA, using a trained router to reduce the number of groups each token must attend to, thereby reducing memory IO during large-batch inference.

A consistent pattern emerges across all of these designs. First, they target *memory-bound* regimes (MHA or GQA), where the bottleneck is KV-cache bandwidth rather than Q-head computation. Consequently, their pruning unit is an entire *group*: pruning a KV head necessarily prunes every Q-head associated with it. Within each retained group, $g_q$ is statically maximized — all Q-heads in the group attend to the KV head, since adding heads is free under memory-bound conditions. Methodologically, four of the five designs rely on trained gates, classifiers, or routers to decide which heads or tokens to activate.

Our work departs from these designs. It targets the *compute-bound* MLA regime, where additional Q-heads incur an extra latency cost (see Section 3.1). Its pruning unit is finer — a single *Q-head within a group* — which allows it to dynamically adjust $g_q$ per group in an *arithmetic-intensity-aware* manner: groups carrying critical tokens retain more Q-heads, while ambient groups operate with fewer. Selection is driven by online importance scores inherited from ArkVale (Chen et al., 2024), keeping the entire pipeline training-free. To our knowledge, TileSparse is the first training-free framework that dynamically adjusts $g_q$ based on hardware arithmetic intensity to reclaim computational resources in compute-bound MLA decoding.

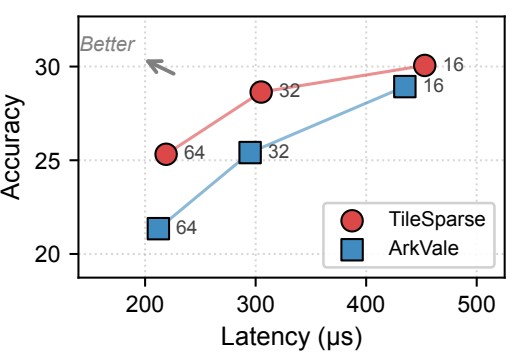

*Figure 14.* Page size ablation on LongBench (GovReport) at a fixed budget of 256 equivalent full-head tokens. Each marker is labelled with its page size. Larger page sizes reduce latency but degrade accuracy on *both* methods, indicating that the accuracy drop is intrinsic to the granularity of page-level K-selection rather than to the intra-page shared Q-head mask of QK 2D Sparsity.

## D. Page Size Ablation

This appendix shows that varying the page size does not compromise our QK 2D Sparsity design. We ablate the page size $\in \{16, 32, 64\}$ on LongBench (GovReport) under a fixed budget of 256 equivalent full-head tokens, comparing TileSparse against the K-only baseline ArkVale on the same axes. Figure 14 reveals two consistent trends. First, increasing the page size from 16 to 64 shifts both curves toward the lower-right corner — latency decreases while accuracy drops because a fixed budget covers fewer pages — and this trend also holds on ArkVale, whose K-only sparsity imposes no Q-head mask, so the accuracy degradation is an intrinsic property of page-level K-selection rather than an artifact of QK 2D Sparsity's intra-page shared mask. Second, the TileSparse accuracy–latency frontier lies strictly above and to the left of ArkVale's, reflecting the effectiveness of QK 2D Sparsity.

