# OpenReview forum: "TileSparse: Arithmetic-Intensity-Aware Sparse Attention for Compute-Bound LLM Decoding"
_ICML.cc/2026/Conference — ICML 2026 regular_

### Official Review · Reviewer_juCL · 2026-02-28

**Soundness:** 2
**Presentation:** 2
**Significance:** 2
**Originality:** 2
**Overall Recommendation:** 4
**Confidence:** 4

**Summary:**

This paper studies sparse attention under the MLA setting, where the arithmetic intensity already exceeds the roofline threshold and enters the compute-bound regime. In such scenarios, reducing memory traffic alone is insufficient, and it becomes meaningful to reduce arithmetic intensity in order to decrease overall computational overhead. Under MLA, since the number of K heads is one, the number of Q heads becomes the dominant contributor to computation. This paper investigates the relationship between arithmetic intensity, MLA, and Q-head computation, and proposes a sparsity mechanism that dynamically allocates the number of tokens computed for each Q head. The method can be interpreted as a Q-head-wise sparsity scheme guided by cumulative attention mass.

**Compliance With Llm Reviewing Policy:**

Affirmed.

**Final Justification:**

I think the rebuttal addressed my main concerns.

**Key Questions For Authors:**

- Please re-explain Figure 4.
- Please clarify the difference between TileSparse and other head-wise attention sparsity based on accumulative attention score.

**Limitations:**

Yes

**Strengths And Weaknesses:**

## strengths

- The motivation is well articulated.

- The emphasis on arithmetic intensity as a guiding principle is insightful. Considering roofline behavior when designing sparsity mechanisms is important and often overlooked in attention optimization work.

## Weakness

- The overall motivation–solution structure is clear, but key technical components are confusing. For example, Figure 4, the most important figure in this paper, is difficult to interpret. The cumulative attention visualization (light color = 100%) seems to contradict its texts (deep color is more important). In addition, Equation (2) and the “page selects heads” formulation obscure what appears to be a straightforward Q-head-wise sparsity method.

- The core idea resembles cumulative-attention-based head sparsity. It is unclear how fundamentally different this is from existing Q-head-wise sparsity methods. Besides, ArkVale is the only baseline; comparisons with other head-wise or 2D sparsity approaches would strengthen the claim.

- The latency gain (Figure 9) appears marginal, and the accuracy improvement (Figures 5, 6, 7), when approaching the performance of full attention, is limited compared to ArkVale. While TileSparse outperforms ArkVale when cache sizes are small, both methods perform significantly worse than full attention under such conditions. Overall, the gains seem modest, particularly given that the proposed method closely resembles Q-head-wise sparsity based on cumulative attention.

---

> ### Author Rebuttal · Authors · 2026-03-31
>
> We thank the reviewer for the feedback and address the concerns below.
>
> **Comment 1 (Fig 4 Re-Explanation):**
>
> > Figure 4, the most important figure in this paper, is difficult to interpret. The cumulative attention visualization (light color = 100%) seems to contradict its texts (deep color is more important).
>
> We appreciate this feedback. The confusion arises from the distinction between "cumulative 100%" and "the last 1%": the light-colored region does not represent 100% of attention mass, but rather the final marginal slice (99%→100%) of cumulative mass — i.e., the sparse long tail with negligible importance. Darker blue, by contrast, captures the core attention mass where importance density is high. We will revise the caption and add boundary annotations to make this distinction visually unambiguous. This long-tail insight is central to TileSparse: pruning these low-value entries reclaims substantial compute budget with minimal accuracy loss.
>
> **Comment 2 (Comparison with head-wise designs):**
>
> > The core idea resembles cumulative-attention-based head sparsity. It is unclear how fundamentally different this is from existing Q-head-wise sparsity methods.
>
> We have expanded our discussion to include recent head-wise sparsity designs: Quest[^a1], DuoAttention[^a2], MoH[^a4], Polar Sparsity[^a5], and LycheeDecode[^a3]. A detailed architectural comparison is provided in our response to Reviewer AeWM (Comment 2). The key distinction is that all these methods operate in memory-bound settings and statically maximize the QK ratio, whereas TileSparse dynamically adjusts the QK ratio based on hardware arithmetic intensity. Methodologically, they all prune heads at the *group* level (a KV head and all its associated Q heads), while TileSparse operates at the finer *Q-head-in-group* granularity to enable adaptive arithmetic intensity control.
>
> **Comment 3 (Baseline expansion):**
>
> > ArkVale is the only baseline; comparisons with other head-wise or 2D sparsity approaches would strengthen the claim.
>
> We have expanded the baselines to include Quest and StreamingLLM (as a proxy for DuoAttention on MLA), and TileSparse consistently outperforms both. Please see our response to Reviewer AeWM (Comment 1) for the detailed results. Regarding head-wise designs (MoH, LycheeDecode, Polar Sparsity): they prune at the group level (KV head + all associated Q heads), which is inapplicable to MLA's single-KV-head structure — group pruning either retains full attention or discards it entirely. Furthermore, they do not change the QK ratio, so even conceptually they cannot improve budget efficiency in compute-bound settings. They also belong to a different category — relying on trained gates or routers — whereas TileSparse is training-free.
>
> **Comment 4 (Latency and accuracy gains):**
>
> > The latency gain (Figure 9) appears marginal
>
> We implemented a custom CUDA kernel for the Select phase to replace the original PyTorch implementation, doubling the wall-time improvement on LongBench (9.9% → 19.8%). Please see our response to Reviewer MRuY (Comment 1) for the full analysis. This demonstrates that better implementation can push actual latency gains toward the theoretical upper bound. Further room remains (e.g., a fully fused end-to-end pipeline), which can be left as future work.
>
> > the accuracy improvement, when approaching the performance of full attention, is limited compared to ArkVale.
>
> Under the same budget, TileSparse consistently achieves higher accuracy than baselines, which translates into reaching a target precision with significantly less budget. For example, on LongBench (Fig 7a), although both methods converge to within 2% of full attention beyond 512 tokens, under a strict 99%-of-full-attention constraint, TileSparse achieves this target with 40% less budget than ArkVale (Fig 8b).
>
> > While TileSparse outperforms ArkVale when cache sizes are small, both methods perform significantly worse than full attention under such conditions. Overall, the gains seem modest.
>
> We agree that under very small cache sizes both methods fall noticeably short of full attention, and such extreme scenarios may be uncommon in practice. However, the budget reduction and updated latency gains enabled by TileSparse's higher accuracy at moderate-to-large budgets are far from modest.
>
> **Comment 5 (Eq 2 "Page Selects Heads"):**
>
> > Equation (2) and the "page selects heads" formulation obscure what appears to be a straightforward Q-head-wise sparsity method.
>
> The "page selects heads" formulation in Eq 2 is intended to ensure efficiency saturation. By modeling sparsity at the page level, we ensure each page's computation stays above the hardware ridge point. A straightforward head-wise approach could result in head counts that fall below the ridge, wasting hardware resources without improving information recall.

---

> > ### Author Rebuttal · Reviewer_juCL · 2026-04-03
> >
> > Thanks for your explanation, and I'll revise my score.

---

> > > ### Author Response · Authors · 2026-04-05
> > >
> > > We thank Reviewer juCL for the detailed comments on figure clarity and design positioning, and for acknowledging our responses.

---

### Official Review · Reviewer_AeWM · 2026-03-03

**Soundness:** 3
**Presentation:** 3
**Significance:** 2
**Originality:** 2
**Overall Recommendation:** 4
**Confidence:** 4

**Summary:**

The paper first use theoretical analysis and profiling to prove that MLA is compute bound. So in this scenario, the sparse attention kernel should can not only focus on sparse KV, but also sparse Q. The author found that in the GQA mode of MLA, a group of query head has sparse attention score related to a same KV token. On the other hand, computing less Q heads can compute more KV tokens within the same amount of time.
For simplicity, the author select $\alpha$ most important KV page to compute all Q heads and $\beta$ less important KV page to compute only $h$ Q heads. The implementation also use an auto-tuner to tune these hyper-parameters.

**Compliance With Llm Reviewing Policy:**

Affirmed.

**Final Justification:**

Concerns are addressed, raise my score from 3->4.

**Key Questions For Authors:**

See above.

**Limitations:**

yes

**Strengths And Weaknesses:**

1. The theoretical analysis of 'compute bound' is well supported by the profiling results. Which forms the foundation of the method.
2. The presentation is clear, especially for Figure 1. The two-tier algorithm is also simple to understand and easy for implementation.
3. The only baseline is ArkVale. More baselines should be added.
4. Lack discussion and performance comparison with other Sparse Head related methods such as [1][2].
5. Typos: in Line 198, 'ArkValue' should be 'ArkVale'.
6. The references are too long. The author should use et.al to make the references clean.

[1] Xiao, Guangxuan, et al. "Duoattention: Efficient long-context llm inference with retrieval and streaming heads." arXiv preprint arXiv:2410.10819 (2024).
[2] Tang, Jiaming, et al. "Quest: Query-aware sparsity for efficient long-context llm inference." arXiv preprint arXiv:2406.10774 (2024).

---

> ### Author Rebuttal · Authors · 2026-03-31
>
> We thank the reviewer for the feedback and address the concerns below.
>
> **Comment 1 (Comparison with Expanded Baselines):**
>
> > The only baseline is ArkVale. More baselines should be added.
>
> Following reviewers' suggestions, we have incorporated additional baselines (including DuoAttention[^a2] and Quest[^a1]). Since DuoAttention's mechanism (splitting KV heads into "retrieval" and "streaming") is incompatible with MLA's single KV head—where a retrieval head would simply revert to full attention—we use StreamingLLM[^a6] to represent the fixed-window streaming behavior. We also included Quest for a comprehensive dynamic sparsity comparison.
>
> Updated results on the accuracy and budget efficiency are available at [Fig. Expanded Baselines](https://anonymous.4open.science/r/ICML26-TS-Rebuttal-Figures-D5E4/more_baselines.png), demonstrating that TileSparse consistently outperforms these baselines. The inferior performance of these baselines is attributed to:
>
> 1. StreamingLLM (DuoAttention proxy for MLA) underperforms as it lacks query-dependent dynamic selection, missing critical long-range semantic anchors.
> 2. Quest's performance is limited by its proxy scoring, which relies on token-level extreme values (cuboid-max). In contrast, the cuboid-mean approach adopted in ArkVale allows us to more accurately evaluate the aggregate semantic importance of an entire page, leading to superior retrieval precision.
>
> **Comment 2 (Comparison with Recent Head-wise Sparsity):**
>
> > Lack discussion and performance comparison with other Sparse Head related methods such as DuoAttention, Quest.
>
> Following the reviewer's suggestion, we have expanded our discussion to include DuoAttention[^a2], Quest[^a1], and additional head-wise sparsity works (MoH[^a4], Polar Sparsity[^a5], LycheeDecode[^a3]). A key observation is that all these methods target memory-bound scenarios and perform head-wise pruning at the *group* level — pruning an entire KV head together with all its associated Q heads — while statically maximizing the QK ratio within each group. In contrast, TileSparse addresses compute-bound MLA inference by pruning at a finer *Q-head-in-group* granularity, dynamically adjusting the QK ratio per group to achieve arithmetic-intensity-aware budget allocation. Additionally, existing head-wise designs rely on training to learn classifiers or routers, whereas TileSparse is entirely training-free.
>
> Key related works:
>
> - **Quest[^a1]:** Performs query-aware K-sparsity by selecting important KV tokens based on token-level extreme values (cuboid-max).
> - **DuoAttention[^a2]:** Partitions heads into "retrieval heads" with full KV cache and "streaming heads" with constant-size cache via gate optimization.
> - **LycheeDecode[^a3]:** Utilizes a HardKuma-based mechanism to partition attention heads into retrieval heads for identifying crucial tokens and sparse heads for reusing those identified tokens.
> - **MoH[^a4] (Mixture-of-Heads):** Treats individual KV heads as experts and utilizes a router to activate Top-K KV heads for each token, thus reducing KV cache fetching.
> - **Polar Sparsity[^a5]:** Employs Selective Group Attention for GQA to reduce memory IO during large-batch inference through a trained router.
>
> The fundamental distinctions of TileSparse are summarized in the following architectural comparison table: [Fig. Head-wise Comparison](https://anonymous.4open.science/r/ICML26-TS-Rebuttal-Figures-D5E4/head-wise-designs.png).
>
> Unlike prior works that statically maximize the number of query heads per key fetch or require specialized training, TileSparse is the first training-free framework that dynamically adjusts the QK ratio based on hardware arithmetic intensity to reclaim computational resources.
>
> [^a1]: Tang et al., *Quest: Query-Aware Sparsity for Efficient Long-Context LLM Inference*, ICML 2024.
> [^a2]: Xiao et al., *DuoAttention: Efficient Long-Context LLM Inference with Retrieval and Streaming Heads*, ICLR 2025.
> [^a3]: Lin et al., *LycheeDecode: Accelerating Long-Context LLM Inference via Hybrid-Head Sparse Decoding*, ICLR 2026.
> [^a4]: Jin et al., *MoH: Multi-Head Attention as Mixture-of-Head Attention*, ICML 2025.
> [^a5]: Shrestha et al., *Polar Sparsity: High Throughput Batched LLM Inferencing with Scalable Contextual Sparsity*, NeurIPS 2025.
> [^a6]: Xiao et al., *Efficient Streaming Language Models with Attention Sinks*, ICLR 2024.
>
> **Comment 3 (References):**
>
> > The references are too long. The author should use et.al to make the references clean.
>
> Thank you for pointing this out. We will use et al. to prune the reference list and focus on core citations to improve readability and save space.
>
> **Comment 4 (Typos):**
>
> > Typos: in Line 198, 'ArkValue' should be 'ArkVale'.
>
> We will fix this and perform a thorough proofread to ensure terminology consistency.

---

> > ### Author Rebuttal · Reviewer_AeWM · 2026-04-03
> >
> > Raise to 4 weak accept

---

> > > ### Author Response · Authors · 2026-04-05
> > >
> > > We thank Reviewer AeWM for the constructive suggestions on baseline expansion and head-wise sparsity comparison, and for recognizing our rebuttal efforts.

---

### Official Review · Reviewer_MRuY · 2026-03-12

**Soundness:** 3
**Presentation:** 3
**Significance:** 3
**Originality:** 3
**Overall Recommendation:** 5
**Confidence:** 3

**Summary:**

The paper addresses a critical paradigm shift in LLM decoding: the transition from memory bound to compute bound operations due to modern architectures like MLA and MTP. The authors identify that traditional K-only sparse attention methods are inefficient in this new regime because they compute all Query heads for every retrieved Key, wasting massive computational resources.
To solve this, the authors propose TileSparse, an arithmetic-intensity-aware sparse attention framework. It introduces QK 2D Sparse, which selectively prunes unnecessary Q-head computations for retrieved keys and reallocates the saved compute budget to fetch more semantically important tokens. Because Q head sparsity exhibits a contour-like pattern where important tokens require more heads than ambient tokens , they propose a Tiered QK 2D Sparse algorithm to balance this. Finally, an online AutoTuner dynamically optimizes the sparsity pattern during runtime. Evaluated on DeepSeek-V3 across RULER and LongBench.

**Compliance With Llm Reviewing Policy:**

Affirmed.

**Key Questions For Authors:**

1.While the theoretical FLOPs reduction is strong, the end-to-end latency gains are currently limited by PyTorch overhead in the Select phase. It would be much more convincing if the authors could implement and evaluate their own fused CUDA kernel to demonstrate the actual hardware speedup and address potential uncoalesced memory access issues caused by dynamic 2D masking.

2.The AutoTuner is triggered every 10k decoding steps. How does this specific static interval perform under highly variable batch sizes or rapidly changing semantic contexts in a real-world serving environment?

3.In Section 5.1, the authors set the page size to 32, and Section 2.2 justifies block-level selection for memory coalescing. However, forcing 32 potentially diverse tokens to share the exact same dynamic Q-head mask may artificially restrict the model's expressiveness. There is no ablation study on the impact of different page sizes. How does altering the page size (e.g., 16, 32, 64) affect the accuracy-efficiency trade-off specifically for the proposed QK 2D Sparse mechanism?

**Limitations:**

yes

**Strengths And Weaknesses:**

Soundness
Strengths: The theoretical foundation is rigorous. The authors effectively utilize the Roofline model to mathematically justify the shift from memory bound to compute bound decoding regimes in MLA+MTP architectures. The observation of Q head sparsity is empirically sound and well-visualized. The experimental setup is robust, comparing against a strong K-only sparsity baseline (ArkVale) over established benchmarks (RULER, LongBench). Ablation studies clearly validate the necessity of the tiered design and the AutoTuner.
Weaknesses: While the theoretical compute reduction is 18.7% and 40.8%, the end-to-end wall-clock latency improvement over ArkVale is relatively modest (6.3% to 9.9%). The authors transparently attribute this gap to PyTorch overhead in the Select phase, but the lack of a fully optimized custom CUDA kernel means the practical utility is currently limited by framework constraints.

Presentation
Strengths: The paper is exceptionally well-written, logically structured, and easy to follow. Figure 1 brilliantly summarizes the conceptual difference between a.i.-agnostic and a.i.-aware sparsity. Figure 4 effectively illustrates the "contour-like" distribution of attention weights across Q-heads, making the motivation for a tiered approach intuitive.
Weaknesses: The algorithmic description relies heavily on understanding the ArkVale proxy scoring mechanism. A brief mathematical summary of the bounding-cuboid proxy formulation within the text itself would make the paper entirely self-contained.

Significance
Strengths: The significance of this work is very high. With frontier models like DeepSeek-V3 / Kimi-K2 / GLM heavily utilizing MLA and MTP to achieve state-of-the-art performance, the community urgently needs sparse attention methods tailored for compute-bound settings. TileSparse directly addresses an immediate and highly relevant bottleneck in LLM serving.

Originality
Strengths: While sparse attention and static head-pruning are well-explored, the synthesis of dynamic, instance-wise 2D QK sparsity applied specifically to MLA's shared latent keys is highly novel. By fundamentally shifting the cost evaluation metric from "memory accessed" to "active compute entries", the authors provide a fresh and necessary perspective for future hardware-aware LLM optimization.

---

> ### Author Rebuttal · Authors · 2026-03-31
>
> We thank the reviewer for the feedback and address the concerns below.
>
> **Comment 1 (Latency Analysis and Optimization):**
>
> > It would be much more convincing if the authors could implement and evaluate their own fused CUDA kernel to demonstrate the actual hardware speedup and address potential uncoalesced memory access issues caused by dynamic 2D masking.
>
> To address concerns regarding the gap between theoretical gains and wall-time latency, we have further optimized the Select phase using a custom CUDA implementation. This targeted optimization significantly expands our advantage in wall-time latency, as illustrated in [Fig. Latency Comparison](https://anonymous.4open.science/r/ICML26-TS-Rebuttal-Figures-D5E4/cuda-select.png).
>
> The CUDA-based Select optimization doubles the wall-time improvement on the LongBench dataset, increasing the gain from 9.9% to 19.8%. Similarly, on Ruler, the optimized wall-time gain is pushed much closer to the upper bound, proving that the previously observed overhead was a localized engineering bottleneck rather than an algorithmic limitation.
>
> This demonstrates that a more refined implementation can successfully push wall-time latency gains closer to theoretical benefits by effectively avoiding engineering/framework overheads. Although the current implementation has already yielded substantial practical gains, a fully fused end-to-end pipeline implementation is reserved for future work.
>
> **Comment 2 (AutoTuner Robustness):**
>
> > The AutoTuner is triggered every 10k decoding steps. How does this specific static interval perform under highly variable batch sizes or rapidly changing semantic contexts in a real-world serving environment?
>
> 10k is a conservative, low-overhead default. For highly dynamic workloads, the interval can be set much lower. Since the AutoTuner approximately doubles the decode latency of the step it activates on, even at an interval of 600 steps (~every 30 seconds), the overhead is only 1/600 (0.17%) of total compute. We will add sensitivity results in the appendix.
>
> **Comment 3 (Page Size Ablation):**
>
> > In Section 5.1, the authors set the page size to 32, and Section 2.2 justifies block-level selection for memory coalescing. However, forcing 32 potentially diverse tokens to share the exact same dynamic Q-head mask may artificially restrict the model's expressiveness. There is no ablation study on the impact of different page sizes. How does altering the page size (e.g., 16, 32, 64) affect the accuracy-efficiency trade-off specifically for the proposed QK 2D Sparse mechanism?
>
> We conducted the requested page size ablation (16, 32, 64) on LongBench GovReport with a fixed budget of 256 equivalent tokens. The results are shown at [Fig. Page Size Ablation](https://anonymous.4open.science/r/ICML26-TS-Rebuttal-Figures-D5E4/page-size-tradeoff.png), revealing a clear accuracy–latency trade-off: larger page sizes yield lower latency but lower accuracy. Importantly, this accuracy drop is not caused by QK 2D's intra-page constraint (forcing the same Q-head set within a page), but is an inherent limitation of 1D K-only sparsity at page granularity — larger pages reduce the number of selectable pages under a fixed budget, lowering recall rate. We verify this by showing that ArkVale exhibits the same trend, confirming the degradation is not specific to QK 2D sparsity. Furthermore, TileSparse consistently achieves a better accuracy–latency trade-off, because pruning Q-heads frees budget to fetch more KV pages, improving recall rate under the same latency constraint.
>
> **Comment 4 (Self-containment):**
>
> > The algorithmic description relies heavily on understanding the ArkVale proxy scoring mechanism. A brief mathematical summary of the bounding-cuboid proxy formulation within the text itself would make the paper entirely self-contained.
>
> We will strengthen the explanation of the ArkVale mechanism, specifically the bounding-cuboid proxy formulation, in the revision to ensure the paper is self-contained without requiring external references to understand the core logic.

---

> > ### Author Rebuttal · Reviewer_MRuY · 2026-04-03
> >
> > Thank the authors for their thorough rebuttal. They have convincingly addressed all my concerns, including the practical latency gap with custom CUDA optimization, the robustness of the AutoTuner under dynamic workloads, and the page size ablation study. The paper remains technically solid, I continue to recommend Accept.

---

> > > ### Author Response · Authors · 2026-04-05
> > >
> > > We sincerely thank Reviewer MRuY for the thorough evaluation and valuable suggestions on latency analysis and page size ablation, which have strengthened the empirical grounding of our work.

---

### Official Review · Reviewer_F5jh · 2026-03-14

**Soundness:** 3
**Presentation:** 2
**Significance:** 3
**Originality:** 3
**Overall Recommendation:** 4
**Confidence:** 3

**Summary:**

The paper provides a sparse attention implementation for inference workloads in which the attention is no longer memory bound due to increased compute cost of multi-token prediction and multi-latent attention. In addition to pruning KV entries, queries are also pruned to reduce compute cost, with non-uniform pruning

**Compliance With Llm Reviewing Policy:**

Affirmed.

**Final Justification:**

I maintain my rating of weak accept. The main limitation of the paper is the significant gap between theoretical and actual speed improvements, even when implemented on the "easiest" GPU (the A100).

**Key Questions For Authors:**

The paper focuses on A100 GPUs. It would be nice to at least provide the arithmetic-intensity estimates for newer GPUs; e.g. would MLA on B200 to be enough to go to the compute-bound regime, or would it only become compute-bound with MLA+MTP.

Two-tiered pruning:
Given that we have an estimator for (Q, KV) pairwise relevance, why do we need the two-tiered approach? E.g., if you first selected KV heads with $\tau_{hw} m_{kv}/2$ queries, and then greedily removed the least important KV and redistributed to the most important Q?

**Limitations:**

The actual implementation provided in the paper does not achieve the promised theoretical speed-up; the paper is up-front about this.

**Strengths And Weaknesses:**

Soundness:
The problem analysis and proposed solution appear sound. The two-tiered approach seems to be a purely empirical choice.

Presentation:
It is hard to see what Figure 3 is trying to show.
I don't understand what Figure 10 is showing. Each individual curve is for a specific head; how does that map to accuracy? Is this pruning in a single layer only?

Significance:
The problem of speeding-up long-context attention for more arithmetic-intensive attention formulations is very relevant. Given that large number of existing A100s, making them more able to run modern inference workloads is helpful particularly to non-gpu rich communities. Still, it would be nice to at least have a discussion of the same problem on more modern hardware.

Originality:
The proposed method is mainly an extension of existing pruning approaches.

---

> ### Author Rebuttal · Authors · 2026-03-31
>
> We thank the reviewer for the feedback and address the concerns below.
>
> **Comment 1 (Hardware Generalizability & B200):**
>
> > The paper focuses on A100 GPUs. It would be nice to at least provide the arithmetic-intensity estimates for newer GPUs; e.g. would MLA on B200 be enough to go to the compute-bound regime, or would it only become compute-bound with MLA+MTP.
>
> We will add a discussion of newer GPUs such as B200 in the revision. As shown in [Fig. B200 Roofline](https://anonymous.4open.science/r/ICML26-TS-Rebuttal-Figures-D5E4/B200.png), the B200 has a hardware arithmetic intensity (a.i.) threshold of 292. Standard MLA (a.i.=256) falls below this threshold and remains memory-bound. However, MLA+MTP2 (a.i.=512) exceeds it and shifts to compute-bound. Thus, TileSparse remains highly relevant for next-generation hardware and scaling strategies.
>
> **Comment 2 (Two-tier vs. Optimal Redistribution):**
>
> > Given that we have an estimator for (Q, KV) pairwise relevance, why do we need the two-tiered approach? E.g., if you first selected KV heads with τ_hw · m_kv / 2 queries, and then greedily removed the least important KV and redistributed to the most important Q?
>
> While greedy redistribution is theoretically optimal (following the weight contours exactly), it is engineering-prohibitive. When launching GPU kernels, the parallelism (Q-head count) must be determined at launch time. Ensuring each tier uses a consistent Q-head count allows the kernel to pre-allocate SM resources and maintain high occupancy. Our two-tier design is an empirical bridge between algorithmic optimality and GPU execution efficiency.
>
> **Comment 3 (Fig 3 Roofline):**
>
> > It is hard to see what Figure 3 is trying to show.
>
> Fig 3 empirically validates the Roofline analysis in Fig 2 by profiling the actual execution time of different attention mechanisms across GPUs. For MHA/GQA, which reside in the memory-bound region, varying the QK ratio (a.i.) does not change the operator execution time. In contrast, MLA operates in the compute-bound region, where the QK ratio (a.i.) directly governs execution time — motivating TileSparse to dynamically control the QK ratio (a.i.-aware) to improve budget efficiency in compute-bound scenarios. We will revise the caption and annotations of Fig 3 in the revision to make this message clearer.
>
> **Comment 4 (Fig 10 Interpretation):**
>
> > I don't understand what Figure 10 is showing. Each individual curve is for a specific head; how does that map to accuracy? Is this pruning in a single layer only?
>
> Each point in Fig 10 represents the accuracy of a single static parameter configuration applied uniformly across all layers. Specifically, the x-axis denotes the number of Critical Tokens, the curve it belongs to indicates the Q-head count in the Ambient Tier, and the y-axis shows the resulting accuracy under that configuration. This ablation validates the QK 2D design by showing how different (Critical Token count, Ambient Q-head count) combinations affect global accuracy under a fixed budget. We will rewrite the caption and add clearer axis annotations in the revision to make this reading guide explicit.

---

> > ### Author Rebuttal · Reviewer_F5jh · 2026-04-04
> >
> > Thanks for the updated figure with the B200 numbers. It shows the technique remains relevant (though I suspect the actual engineering challenge of turning theoretical speed-ups into practical ones to be significantly harder on B200 than on A100)
> >
> > Figure 3 is also simply too small. There's something happening in the lower left corner; the roofline behaviour, i.e., constant followed by an increase once the threshold is crossed, is not visible. The R² value is for a linear fit? Isn't the function supposed to be piecewise linear?
> >
> > Variable size (e..g, work-stealing) kernels _can_ be implemented. However, I agree that implementation difficulty is a sufficient argument why not to do this for this paper; it just should be discussed explicitly.

---

> > > ### Author Response · Authors · 2026-04-05
> > >
> > > We thank Reviewer F5jh for the continued engagement and the constructive follow-up. We address each point below.
> > >
> > > **Follow-up 1 (Figure 3 Readability & Piecewise Linearity):**
> > >
> > > > Figure 3 is also simply too small. There's something happening in the lower left corner; the roofline behaviour, i.e., constant followed by an increase once the threshold is crossed, is not visible.
> > >
> > > Thank you for pointing out the lack of clarity in the lower-left (memory-bound) region. We have revised Fig 3 by adding a magnified inset to clearly show the memory-bound latency plateau: [Fig. 3 Revised with Inset](https://anonymous.4open.science/r/ICML26-TS-Rebuttal-Figures-D5E4/attn_roofline_model.png).
> > >
> > > > The R² value is for a linear fit? Isn't the function supposed to be piecewise linear?
> > >
> > > Yes, the function is piecewise linear — constant in the memory-bound region and linearly increasing in the compute-bound region. The R² is fitted exclusively over the compute-bound segment. We will emphasize this scope in the revised caption.
> > >
> > > **Follow-up 2 (Two-Tier Design & Work-Stealing Discussion):**
> > >
> > > > Variable size (e.g., work-stealing) kernels can be implemented. However, I agree that implementation difficulty is a sufficient argument why not to do this for this paper; it just should be discussed explicitly.
> > >
> > > We agree. We will add an explicit discussion of the two-tier design in the revision, clarifying the trade-off between theoretical optimality and implementation complexity. Thank you for suggesting variable-size (work-stealing) kernels — we will also discuss this as a promising future direction.
> > >
> > > **Follow-up 3 (B200 Engineering Challenges):**
> > >
> > > > Thanks for the updated figure with the B200 numbers. It shows the technique remains relevant (though I suspect the actual engineering challenge of turning theoretical speed-ups into practical ones to be significantly harder on B200 than on A100).
> > >
> > > We agree that newer hardware such as B200 places higher demands on implementation quality to translate theoretical speedups into practical gains. For instance, B200 has significantly more SMs than A100, which amplifies kernel tail effects — without careful scheduling, idle SMs waiting for the slowest to finish leads to greater resource waste.

---

### Decision · Program_Chairs · 2026-04-30

**Decision:**

Accept (regular)

**Comment:**

This paper proposes TileSparse, a sparse-attention scheme tailored to the compute-bound decoding regime that arises with MLA and MTP. Starting from a Roofline analysis showing that MLA decoding is compute-bound rather than memory-bound, the paper introduces an arithmetic-intensity-aware cost model and a 2D sparsity pattern that prunes Q-head/key interactions in addition to keys, plus a tiered variant and an online AutoTuner. Experiments on DeepSeek-V3 with RULER and LongBench show consistent accuracy gains over strong dynamic K-only baselines under tight budgets, and meaningful end-to-end latency improvements once a custom CUDA Select kernel is used.

All reviews recommend acceptance. Reviewers consistently praised the timeliness and significance of targeting compute-bound MLA/MTP decoding (which is highly relevant to current frontier models such as DeepSeek-V3 and Kimi), the rigor of the Roofline-based motivation, the novelty of 2D Q-K sparsity for MLA's shared latent keys, and the clarity of the core figures motivating the tiered design. The author response added baselines (Quest, StreamingLLM as a DuoAttention proxy), and a custom CUDA kernel that roughly doubles the wall-clock latency improvement on LongBench.

A residual limitation is that the implementation does not fully realize the theoretical compute savings end-to-end; the paper is upfront about this and the additional CUDA optimization narrows the gap, especially on modern GPUs like B200. I share the reviewers' view that the conceptual contribution and empirical evidence are strong enough to outweigh this engineering gap, especially given the clear practical relevance.

Overall, the paper is technically sound, well-motivated, and addresses an immediate bottleneck in modern LLM serving. I recommend acceptance.